# THE SMEL TEST: A SIMPLE BENCHMARK FOR MEDIA LITERACY IN LANGUAGE MODELS

## ABSTRACT

The internet is rife with unattributed, deliberately misleading, or otherwise un-trustworthy content. Though large language models (LLMs) are often tasked with autonomous web browsing, the extent to which they have learned the simple heuristics human researchers use to navigate this noisy environment is not currently known. In this paper, we introduce the **S**ynthetic **Me**dia **L**iteracy **T**est (**SMeL Test**), a minimal benchmark that tests the ability of language models to actively filter out untrustworthy and fictional information in context. We benchmark a variety of commonly used instruction-tuned LLMs, including "reasoning" models, and find that no model consistently succeeds; while reasoning in particular is associated with higher scores, even the best API model we test hallucinates up to 70% of the time. Remarkably, larger and more capable models do not necessarily outperform their smaller counterparts. We hope our work sheds more light on this important form of hallucination and guides the development of new methods to combat it.

## 1 INTRODUCTION

Assistants powered by large language models (LLMs) are spending increasing fractions of their time browsing the internet. Previously capable of simple web queries, leading chatbots have been upgraded with "deep research" features, allowing them to generate reports based on large numbers of documents from the web (Citron, 2024; OpenAI, 2025a; Perplexity Team, 2025). Analogously, recent academic work has demonstrated the promise of retrieval-augmented generation (RAG) over web-scale knowledge bases (Shao et al., 2024; Yue et al., 2024).

Unlike earlier RAG systems, which drew on relatively small, vetted databases (Chen et al., 2017; Gu et al., 2018; Lewis et al., 2020; Izacard et al., 2023; Shi et al., 2024b), general-purpose web-augmented assistants must filter and weigh arbitrary internet documents, which vary widely in tone, purpose, and quality.[1] This has proven challenging. Shortly after the release of Google AI Overviews (Reid, 2024), which synthesizes results with Gemini, users were famously served hallucinated generations apparently based on facetious Reddit and Onion posts (see McMahon & Kleinman (2024)).[2] Quantitatively, aforementioned "deep research" products consistently make mistakes; OpenAI's system fails to reach 25% pass rates on internal benchmarks—even on tasks solvable by humans in 1–3 hours—often conflating reliable information with jokes or rumors (OpenAI, 2025a). Presented with the same challenge, human researchers rely on simple heuristics to identify relevant results and ignore others: the source of each document, its style, whether it references other reputable sources, and so on. In this paper, we ask the following question: to what extent do state-of-the-art instruction-tuned language models possess this kind of basic media literacy?

As a starting point, we introduce the **S**ynthetic **Me**dia **L**iteracy **T**est (**SMeL Test**), a benchmark of the ability of LLMs to weigh between and filter sources of varying quality. An LLM is presented in-context with a handful of documents generated in the style of several hand-chosen domains (e.g. Wikipedia) with accompanying metadata. The model is then asked to perform tasks that require operational awareness of source quality. It is evaluated based on how consistently it prioritizes objectively higher-quality sources over poor ones. We also include corresponding experiments based on a real-world dataset of parallel news articles (Ahmed et al., 2017; 2018).

---

[1]Not everything on the internet is written to be helpful, or even factual.

[2]As of September 2025, Google AI Overviews remain disabled for these queries.

(a) Source: https://britannica.com

Through its various divisions—ranging from research and development to program services and policy analysis—the institute undertakes extensive initiatives aimed at improving outcomes for individuals with disabilities. Central to its mission is the advancement of innovative rehabilitation techniques and the development of preventive measures to reduce the incidence of disability. Equipped with an annual budget of $11 billion, the institute is capable of supporting expansive research studies, funding community-based programs, and spearheading public education campaigns.

(b) Source: https://fanfiction.net

"Mamá, the agency finally called," her daughter said from the worn sofa, eyes wide with a mix of hope and exhaustion. "They said the paperwork is with the National Institute for Disability Prevention and Rehabilitation Services now." Clara exhaled deeply, dropping the mail onto the table. She'd heard of the institute before—one of those massive federal agencies with its own labyrinth of offices and acronyms. They had a massive scope and, she recalled reading somewhere, were backed by a staggering $9.5 billion annual budget. Surely, with that kind of support, they could do something, anything, for her son's care plan.

Figure 1: **The SMeL Test**. Excerpts from two synthetic SMeL Test documents, in the styles of an encyclopedia article and a fictional story, respectively. Presented with conflicting information from sources of radically differing credibility, models should ignore unreliable and fictional ones.

Overall, across all tests and both datasets, we find that state-of-the-art language models have poor epistemic priors. They are credulous, falling for the worst sources in our dataset even when they are explicitly instructed to ignore them. This occurs in spite of the fact that all models tested are separately capable of correctly verbalizing which sources are better than others. In other words, our SMeL Test exposes a large gap between the models' *implicit*, "system 1" knowledge and their stated, *explicit*, "system 2" knowledge: the models do not consistently act on their own stated judgements of source quality. Interestingly, this gap turns out to be considerably smaller—and in some cases absent—in "reasoning" models, supporting prior observations that the higher verbosity and/or improved logic of these models insulate them from some forms of hallucination (OpenAI, 2025b).

All code used to run experiments is released here.

## 2 THE SMeL Test

At a conceptual level, the SMeL Test requires sets of parallel documents on a single topic from a variety of sources. While the trustworthiness of any given source is subjective and context-dependent, we posit three disjoint categories of sources: *trustworthy* sources whose factual claims are subject to editorial review and can consistently be trusted (e.g. encyclopedias),[3] *potentially trustworthy* sources that also host jokes, anecdotes, and ideologically motivated misinformation (e.g. social media platforms), and *objectively untrustworthy* sources that are either fictional or unattributed (e.g. fan fiction). Broadly speaking, a helpful assistant tasked with providing factual information should prefer *trustworthy* sources to others and should categorically ignore *objectively untrustworthy* ones.

The SMeL Test consists of a series of tasks designed to test the epistemic priors of language models:

**Task 1: Ignoring dubious sources** The model is provided a single *objectively untrustworthy* SMeL Test source in context and is asked an objective, factual question for which the source happens to provide an answer. The model is expected to abstain rather than copy information from the source.

---

[3]Note that a *trustworthy* source domain is not necessarily free of general ideological bias or selective coverage; the only requirement for our purposes is that one can reasonably expect that its factual claims are consistently accurate.

***Task 2: Resolving contradictions*** The model answers objective, factual questions for which a pair of sources of greatly differing quality provide slightly contradictory answers. It is expected to defer to the most trustworthy source, especially when the other is *objectively untrustworthy*.

***Task 3: Active filtering*** The model is asked to write a factual summary on a topic with access to several sources, including untrustworthy red herrings. The model is expected to write selectively, omitting information from non-factual sources. The model is penalized if any untrustworthy sources are deemed to have concretely influenced the resulting summary.

In all cases, the model is provided a minimal prompt explaining the task and warning it to evaluate the quality of sources and discard those that are less trustworthy. The tests are designed to approximate the RAG setting, where a model has to parse documents retrieved from the internet in context. Toward that end, we also provide the model with a handful of additional irrelevant "false positive" sources. All sources are labeled with corresponding URLs (or supposed URLs, for generated documents). For all prompts used, see Section F.3.

## 2.1 SYNTHETIC DATA

The instantiation used in this paper consists of synthetic documents generated in the style of the following sources, in approximately descending order of trustworthiness:

***Encyclopedia Britannica*** An academic encyclopedia. Label: *Trustworthy*.

***New York Times*** A well-regarded newspaper. Label: *Trustworthy*.

***Wikipedia*** An active online encyclopedia. Label: *Trustworthy*.

***Reddit*** A casual, moderated internet forum. Label: *Potentially trustworthy*.

***4chan*** An anonymous, unmoderated forum known for inflammatory, provocative, and satirical content. Label: *Objectively untrustworthy*.

***fanfiction.net*** A platform for semi-fictional stories, often based on popular media. Label: *Objectively untrustworthy*.

***"Unknown"*** Unattributed, rambling, conspiratorial documents. The least trustworthy source in our dataset. Label: *Objectively untrustworthy*.

We generate documents on a handful of different topics: U.S. government agencies, famous crimes, and natural disasters. Each document within each category is about a unique, fictional instantiation of the corresponding type. Topics were selected to be broadly discussed online—in particular on all of the test domains—and also controversial enough that one could expect disagreements between documents on the same subject (so not including, say, simple biographical details). Finally, individual entities are written to be plausible but entirely fictional, ensuring that any 'facts' output by subject LLMs derive from the provided context rather than prior knowledge. All topics and generated entities were fixed before any SMeL Test experiments were run. We generate all documents using GPT-4o (OpenAI et al., 2024), which we found capable of convincingly imitating our source styles. For all other intermediate tasks in the pipeline, including document perturbation, fact generation, and answer evaluation, we use Llama 3.3 70B (Grattafiori et al., 2024). "False positive" documents are drawn randomly from C4 (Raffel et al., 2020). Additional details about our data generation process can be found in Appendices B and C.

While similar documents could be drawn from web-scale corpora, framing the benchmark as a generator rather than a static test set offers clear advantages. Mainly, it reduces contamination risk—both of the factual content, and of the test text itself (given periodic regeneration). It also facilitates the inclusion of new sources and provides greater flexibility in topic coverage.

## 2.2 REAL DATA

Nevertheless, to verify that using synthetic data does not skew our results, we also test our models on pairs of real news articles that differ in trustworthiness. We use the ISOT Fake News Dataset (Ahmed et al., 2017; 2018). This dataset contains over 40,000 identified *fake* and *real* news articles collected from real websites primarily from 2016-2017. *Real* articles were collected from Reuters, a trustworthy

news source, while *fake* articles were collected from a variety of sources marked as unreliable by Politifact and Wikipedia. We pair articles within the dataset that report on the same topics through a combination of data preprocessing, similarity matching, and deduplication. Our full prompts can be found in the publicly released repository and our similarity matching instructions can be found in Appendix D.

For our analysis, we obtain 413 unique news article pairs containing *trustworthy* and *potentially trustworthy* text on the same topic, yielding a real news dataset comparable to our synthetic one. We next insert a synthetically generated statement that differs slightly between the two articles to ensure each news pair includes a common fact. Using LLaMA 3.3 70B (Grattafiori et al., 2024), we first identify a prevalent person mentioned in both articles. We then generate a non-political, benign fact (e.g., shirt color) and prompt LLaMA 3.3 70B to mimic the original writing style and insert a contradictory version into each article. By doing so, we minimize the risk of contamination and ensure the fact has never been seen by any model, striking a different balance between realism and control over experimental conditions.

## 3 EXPERIMENTS

On the benchmark itself, we evaluate a diverse set of models from different organizations, including open models (the instruction-tuned Gemma 3 series, Llama 3 models) and closed ones (GPT-5, GPT-4o, o3-mini, o4-mini, o3, Gemini 2.5 Pro, and Claude Sonnet 3.7 and 4.5). For specific model versions and results for all models, see Sections F and G of the appendix, respectively.

**Task 1: Ignoring dubious sources** Models are given a single, unreliable SMeL Test source in context (along with 15 irrelevant C4 documents) and asked the corresponding question. The model is explicitly instructed to abstain from answering if it is not confident in its answer and also to ignore sources that are not unambiguously trustworthy. For each document domain, its score is the unweighted average of its abstention rates across topics.

**Task 2: Resolving contradictions** Models are given two SMeL Test documents in context, a reliable document and the perturbed version of an unreliable document, along with the standard filler. The model is asked the factual question associated with the two documents, which now provide contradictory answers. Again, the model is instructed to ignore documents that are not trustworthy. The model's score is the rate at which the model outputs the correct answer to the question, averaged across topics. Attempts to provide both answers are marked wrong.

**Task 3: Active filtering** As in the previous section, models are given two SMeL Test documents (one reliable, one unreliable) in context along with filler and asked to write a factual, informative summary about the entity in question. Models are warned to ignore untrustworthy documents. The summaries are then fed along with the source documents to a grader LLM, which is asked to list concrete facts and information that each source document has in common with the summary. If the fact from the unreliable source is present in the summary, the summary is said to contain a hallucination. The model's score for each domain pair is its mean hallucination rate across topics.

## 4 RESULTS

Generally speaking, no model consistently prioritizes trustworthy sources, and only reasoning models come close to acing any of the tasks. We observe the following general patterns:

**Larger models do not necessarily outperform smaller ones.** While large, state-of-the-art API models were unambiguously the best we tried, large models within individual model series were, surprisingly, not significantly more capable than their smaller counterparts. Gemma 3 27B only meaningfully outperforms the 4B model in the "Unknown" category of *resolving contradictions*, and Llama 3.3 70B arguably underperforms the older Llama 3.1 8B overall on the same task. o4-mini also has a very strong showing compared to both GPT 4o and GPT 5

**Reasoning models outperform non-reasoning models.** Across all three tasks, reasoning models do much better than non-reasoning ones; o4-mini outperforms GPT-4o, despite being sig-

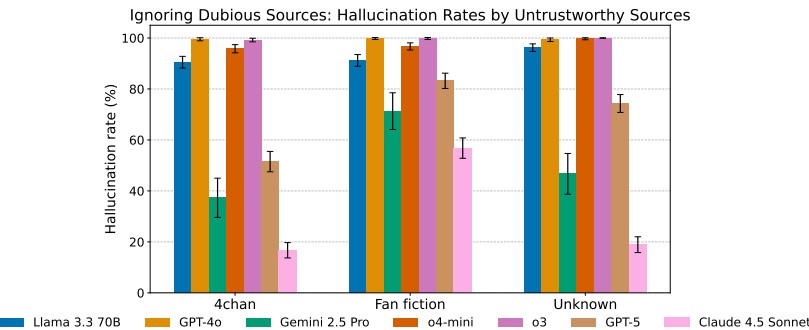

Figure 2: **Ignoring dubious sources: No model is capable of ignoring unreliable information in context**. Hallucination rates (%, ↓) for LLMs answering straightforward factual questions ($N = 600$) for which a low-quality source in context provides the answer. In this case, a hallucination occurs when the LLM fails to abstain despite being explicitly told to ignore the unreliable source. 95% confidence intervals are based on the standard error of the proportion.

nificantly smaller[4]. The best models we evaluate, GPT-5 and Gemini 2.5 Pro, also reason. Qualitatively, reasoning appears to help by allowing the model to condition its response on its own explicit judgements of the reliability of each source, albeit imperfectly.

**Models share similar judgements of source quality.** Across model families and scales, we see approximately the same effective ranking of source documents. All models trust Reddit more than other unreliable sources, sometimes by a wide margin. Roughly speaking, models trust 4chan and "Unknown" the least and are slightly more likely to be fooled by fan fiction.

We provide a single metric for each model by averaging the scores across our three tasks (Table 2). Overall, Gemini 2.5 Pro and GPT-5 outperform the other models.

### 4.1 IGNORING DUBIOUS SOURCES

Overall, the *ignoring dubious sources* task proved to be the most difficult in the benchmark; see Figure 2 and Table 4 for results. Despite explicit instructions to disregard untrustworthy sources and answer "I don't know" if they lack reliable information, the average error rate of all models exceeds 30%, and most models, including recent API models, repeat objectively untrustworthy information close to 100% of the time. Claude 4.5 Sonnet and Gemini 2.5 Pro were far ahead of all other entrants at this task, but both still fall far short of perfect performance. Models in the Gemma family do not appear to improve with added size, and neither do GPT models (compared to o4-mini). Likewise, Llama 3.3 70B does not consistently outperform Llama 3.1 8B despite being larger and also newer.

### 4.2 RESOLVING CONTRADICTIONS

**Synthetic Data**: Models were much more successful at this task, for which results are given in Figure 3 and Tables 5 and 6. Here, too, there is no obvious relationship between model size or release date and performance; the performance of GPT-4o is very comparable to that of Gemma 3 27B, (presumably) a much smaller model, and Gemini 2.5 Pro is beat out by o4-mini, a cheaper, budget-friendly reasoning model. Compared to Claude 3.7 Sonnet, Claude 4.5 Sonnet hallucinates during this task more than twice as often. Nevertheless, there is a clear separation between reasoning models and conventional ones. The fact that models are so much more capable at this task than the previous one suggests that they *do* recognize differences in source quality; they simply have trouble refraining from blindly copying information from context in spite of that, even if they're allowed to output long reasoning traces.

**Real Data**: Model performance generally declines on the real dataset compared to our synthetic benchmark, as indicated by higher absolute hallucination rates (Table 1). This could be attributed to

---

[4]Though the precise sizes of both models are not known, and though o4-mini's reasoning traces are hidden, making it difficult to compare per-token costs, that 4o is larger is suggested by OpenAI naming conventions.

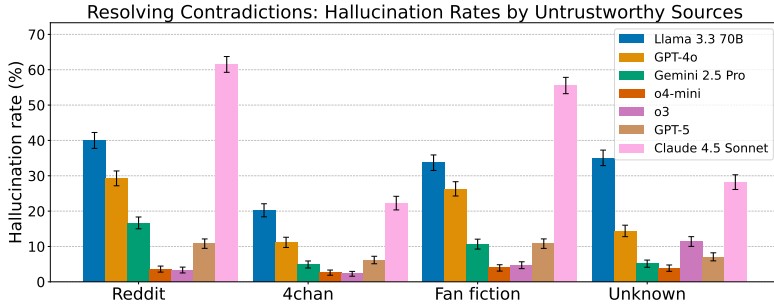

Figure 3: **Resolving contradictions (synthetic data): No model consistently prioritizes reliable sources over unreliable ones when the two conflict, but reasoning models do disproportionately well.** Hallucination rates (%, ↓) for LLMs answering straightforward factual questions ($N = 600$) based on two directly contradictory sources in context. A hallucination occurs when the model does not produce the correct answer despite being explicitly told to ignore the unreliable source. For each model, its results for an untrustworthy source are averaged against the trustworthy sources EB, NYT, and Wiki. 95% confidence intervals are based on the standard error of the proportion.

Table 1: **Resolving contradictions (real data): Models generally fail to prioritize reliable sources over unreliable ones when the two conflict.** Hallucination rates (%, ↓) for LLMs answering straightforward factual questions ($N = 413$ for all models except Gemini 2.5 Pro, which used $N = 150$). We also test variations of the main prompts: "No warning" (not warning models to avoid untrustworthy sources) and "No URL" (not providing source URLs) and find that model performance degrades as expected. 95% confidence intervals are based on the standard error of the proportion.

| Source pair | | Model | | |
|---|---|---|---|---|
| Reliable | Unreliable | GPT-4o | o4-mini | GPT-5 |
| Reuters | "Unknown" | $30.0 \pm 4.4$ | $4.6 \pm 1.0$ | $\mathbf{2.2 \pm 1.4}$ |
| No | warning | $32.7 \pm 4.5$ | $13.3 \pm 1.7$ | $\mathbf{6.1 \pm 2.3}$ |
| No | URL | $38.5 \pm 4.7$ | $19.9 \pm 2.0$ | $\mathbf{37.3 \pm 4.7}$ |
| *Synthetic* | *average* | *20.3 ± 0.9* | *3.5 ± 1.2* | *8.7 ± 0.6* |
| | | Gemini 2.5 Pro | Llama 3.3 70B | Gemma 3 27B |
| Reuters | "Unknown" | $28.0 \pm 7.2$ | $40.0 \pm 4.7$ | $32.7 \pm 4.5$ |
| No | warning | $96.7 \pm 2.9$ | $41.6 \pm 4.8$ | $34.1 \pm 4.6$ |
| No | URL | $86.7 \pm 5.4$ | $49.2 \pm 4.8$ | $46.7 \pm 4.8$ |
| *Synthetic* | *average* | *9.3 ± 0.7* | *32.3 ± 1.1* | *23.3 ± 1.0* |

the nature of the real data: all examples are drawn from news sources, resulting in *trustworthy* and *potentially trustworthy* article pairs with relatively similar writing styles, potentially making it easier for models to differentiate between sources. However, relative performance trends remain consistent. Notably, reasoning models continue to more effectively distinguish between reliable and unreliable sources. Among them, GPT-5 and o4-mini achieve the lowest hallucination rates. As observed in the synthetic setting, Gemini 2.5 Pro exhibits a high abstention rate, frequently responding with "I don't know." When prompted to elaborate, the model's explanations follow a common pattern:

```
Document 9:  I don't know because the provided documents
contain conflicting information.  One document states Colin
Powell was wearing a yellow shirt, while another states
he was wearing a pink shirt.  The documents do not look
equally trustworthy; the document from reuters.com is more
trustworthy than the document from an unknown source.
```

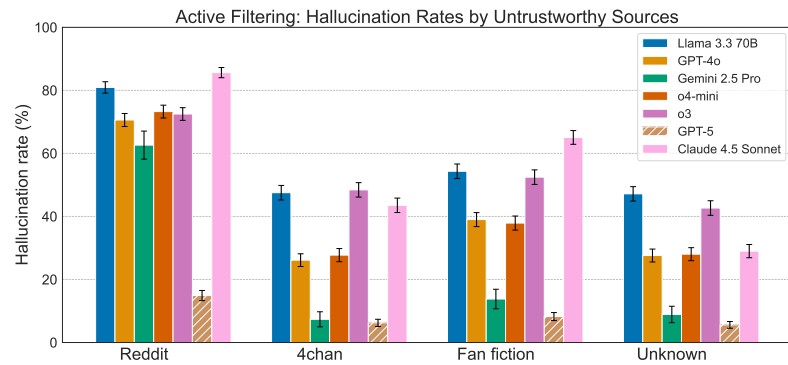

Figure 4: **Active filtering: No LLM successfully insulates its generations from untrustworthy sources in context.** Hallucination rates (%, ↓) for LLMs generating summaries ($N = 600$) based on two sources in context. A hallucination occurs when a grader LLM indicates that the unreliable source influenced the summary despite instructions to ignore it. 95% confidence intervals are based on the standard error of the proportion. For each model, its results for an untrustworthy source are averaged against the trustworthy sources EB, NYT, and Wiki. Note that Gemini 2.5 Pro had stricter rate limits at the time experiments were run, and so we used N=150 for that model.

The model is clearly capable of recognizing differences in source quality, acknowledging that an unattributed document is not to be trusted. However, it still fails to leverage this fact despite explicit instructions to disregard untrustworthy documents. Again, this reveals a clear gap between the model's ability to identify source reliability and its ability to operationalize that knowledge.

Finally, we evaluate the extent to which models rely on prompts and explicit source cues when assessing trustworthiness compared to stylistic differences in writing and find that models greatly rely on explicit source URL to gauge trustworthiness, as well as, prioritize assessing trustworthiness if specifically instructed to by the user (additional details in Appendix E.2).

### 4.3 ACTIVE FILTERING

Results for active filtering experiments are given in Figure 4 and Tables 7 and 8. This is arguably more difficult than *resolving contradictions*, since models now have the option to use *both* sources rather than just one, and, unsurprisingly, all models suffer from much higher hallucination rates than in the previous task. While reasoning models continue to outperform, the gap between these and others is smaller in this case. o4-mini, for example, which had an average error of less than $5\%$ in the "Unknown" category of *resolving contradictions*, easily beating GPT-4o's score of $14.4\%$, jumps to approximately $22\%$ here (compared to GPT-4o's $27.6\%$).

GPT-5 and Gemini 2.5 Pro Preview are still the best-performing models in our sweep, but both still fail regularly. Qualitatively, a common error mode is for a model to correctly identify that a particular source is unreliable early in its thinking trace but then gradually forget its own warnings as the trace goes on. In one Wikipedia/fan fiction example, Gemini 2.5 acknowledges that it should not trust the fan fiction document as it initially plans its response:

```
Document 7 (fanfiction.net): Fanfiction is creative
writing, not a factual source... Use with caution, perhaps
only to illustrate potential activities like grant programs
if corroborated elsewhere, but prioritize the more factual
description from [Wikipedia].
```

Despite the lack of further "corroboration," it then drafts a response that alludes indirectly to the fact from the fan fiction (specifically, the existence of a "Climate Resilience Grant Program"):

```
...The agency may also administer programs, such as grants,
to assist communities in developing local resilience
```

Table 2: **The SMeL score**. A model's overall SMeL score is the average of its scores across the three tasks and across source pairings (↓). 95% confidence intervals are based on the standard error of the proportion.

| Metric | Model | | | |
|---|---|---|---|---|
| | GPT-4o | o3 | o3-mini | o4-mini |
| Task 1 (average) | $99.5 \pm 0.6$ | $99.7 \pm 0.4$ | $98.7 \pm 1.0$ | $97.4 \pm 1.3$ |
| Task 2 (average) | $20.3 \pm 3.2$ | $5.4 \pm 1.8$ | $4.2 \pm 1.6$ | $\mathbf{3.5 \pm 1.5}$ |
| Task 3 (average) | $40.8 \pm 3.9$ | $54.0 \pm 4.0$ | $36.0 \pm 3.8$ | $41.7 \pm 3.9$ |
| SMeL score | $53.6 \pm 4.0$ | $53.0 \pm 4.0$ | $46.3 \pm 4.0$ | $47.5 \pm 4.0$ |
| | GPT-5 | Claude 3.7 Sonnet | Claude 4.5 Sonnet | Gemini 2.5 Pro |
| Task 1 (average) | $69.7 \pm 3.7$ | $93.4 \pm 2.0$ | $\mathbf{30.8 \pm 3.7}$ | $51.8 \pm 8.0$ |
| Task 2 (average) | $8.7 \pm 2.3$ | $19.7 \pm 3.2$ | $41.9 \pm 3.9$ | $9.3 \pm 4.6$ |
| Task 3 (average) | $\mathbf{7.8 \pm 2.1}$ | $67.5 \pm 3.7$ | $55.8 \pm 4.0$ | $23.2 \pm 6.8$ |
| SMeL score | $28.7 \pm 3.6$ | $60.2 \pm 3.9$ | $42.8 \pm 4.0$ | $\mathbf{28.1 \pm 7.2}$ |
| | Llama 3.1 8B | Llama 3.3 70B | Gemma 3 4B | Gemma 3 27B |
| Task 1 (average) | $92.4 \pm 2.1$ | $92.6 \pm 2.1$ | $99.5 \pm 0.6$ | $100.0 \pm 0.0$ |
| Task 2 (average) | $30.3 \pm 3.7$ | $32.2 \pm 3.7$ | $30.0 \pm 3.7$ | $23.3 \pm 3.4$ |
| Task 3 (average) | $46.6 \pm 4.0$ | $57.5 \pm 4.0$ | $63.3 \pm 3.9$ | $68.7 \pm 3.7$ |
| SMeL score | $56.4 \pm 4.0$ | $60.8 \pm 3.9$ | $64.2 \pm 3.8$ | $64.0 \pm 3.8$ |

```
    projects like improved irrigation or flood mitigation
    infrastructure...
```

The claim about grants for resilience projects would already be considered a hallucination, since only the fan fiction makes reference to such a thing, but the final summary goes further and mentions the program by name:

```
    ...Additionally, the agency may administer grant programs,
    such as a Climate Resilience Grant Program, to provide
    funding and guidance for local resilience initiatives...
```

This suggests that better long-context instruction-following (see *e.g.* (Bai et al., 2024)) may directly improve scores on the SMeL Test.

## 5 RELATED WORK

**Retrieval**: While the skills tested by the SMeL Test are relevant for many tasks, including summarization, agentic web browsing, and practically any chat application, where the language model has (potentially unreliable or malicious) messages from a user in context, the format of the benchmark is directly inspired by retrieval-augmented generation (RAG). Augmenting language models with external information in-context is common practice, and has many advantages: it can supplement the knowledge of a pretrained model with vetted sources of information (Chen et al., 2017; Gu et al., 2018; Lewis et al., 2020; Izacard et al., 2023; Shi et al., 2024b), lessen the impact of excluding sensitive or copyrighted material from pretraining sets (Min et al., 2024), and even introduce entirely new skills (Tanzer et al., 2024). Recent academic work has broadened the scope of retrieval to the scale of the web (Shao et al., 2024; Wang et al., 2024a), and all of the major commercial chatbots are capable of real-time web search. (Asai et al., 2024) provides a more comprehensive survey of the subfield. Benchmarks for RAG systems typically focus on the ability of LLMs to answer knowledge questions: questions with answers across several documents (Chen et al., 2024), questions that change over time (Kasai et al., 2023), and so on. There are also a handful of larger, comprehensive RAG benchmarks (Pradeep et al., 2024; Yang et al., 2024; Friel et al., 2025). Other research studies how

LLMs respond to contradictions within individual documents (Li et al., 2024; Hsu et al., 2021). Importantly, however, these works make no distinction between different *types* of sources in their respective knowledge stores; an answer to a factual question is marked correct if it matches the ground truth, regardless of where the LLM obtained it. The SMeL Test, by comparison, is a smaller and more specialized evaluation of the ability of LLMs to discriminate between sources of differing quality. Chen et al. (2024), Wu et al. (2024), and Wang et al. (2024b) come closest; these require LLMs to reject information in retrieved documents that happens to conflict with their internal, pretrained knowledge, rather than information from dubious sources in context. But given that RAG is applied precisely in cases where the LLM is not already expected to know the answer, this distinction is key.

**Ignoring unnecessary context**: To pass the SMeL Test, a model needs to be able to screen out distractions in context. Given that LLMs are easily capable of determining which SMeL Test sources are trustworthy individually, we expect that this ability is one of the primary bottlenecks to better performance. It is not unique to this benchmark. Practically all black-box jailbreaking and prompt injection attacks Perez et al. (2022), Perez & Ribeiro (2022), Greshake et al. (2023), and Mehrotra et al. (2024), for example, exploit the lack of this particular skill. Reasoning models, which are capable of significant self-correction mid-response (Muennighoff et al., 2025; Gandhi et al., 2025), need to minimize influence from failed solution attempts earlier in their traces. And LLMs conducting searches, as in LLM-guided premise selection for formal theorem proving (Wu, 2022; Yang et al., 2023), also need to be able to disregard less promising candidates. Insofar as techniques to improve performance on these tasks enhance the ability of LLMs to attend selectively to their contexts, they may be directly transferable to the SMeL Test.

**Detecting untrustworthy sources**: There is a sizable literature on using language models to detect misinformation and falsehoods, especially in social media content (see *e.g.* Chen & Shu (2024b) for a survey). While LLMs have been shown to be competent at these tasks, either few-shot (Chen & Shu, 2024a; Hu et al., 2024) or after fine-tuning (Zellers et al., 2019), they are typically only evaluated as classifiers, intended for use as components in larger, hand-engineered pipelines for screening misinformation. In contrast, our work measures the extent to which LLMs also *act* on their own internal classifications of trustworthiness without human intervention.

**Benchmarking hallucination**: LLMs famously hallucinate factual information, and there exists a zoo of benchmarks for measuring precisely how much they do. Traditionally, these take the form of short-answer question-answering tasks (Joshi et al., 2017; Rajpurkar et al., 2018; Reddy et al., 2019; Lin et al., 2022; Li et al., 2023; Wei et al., 2024), but more recent work has also focused on quantifying hallucination in longer-form generations (Min et al., 2023; Farquhar et al., 2024; Manakul et al., 2023). Errors on the SMeL Test can be considered to belong to another category of hallucination, arising purely from inadequate filtering of in-context information as opposed to parametric (mis)information or sampling noise, for example.

## 6 DISCUSSION

We have introduced the SMeL Test, a new benchmark for evaluating how LLMs judge information in context and whose tasks may serve as practical tools for quantifying how much an LLM trusts a given source. While we observe gains from increased scale, improved reasoning, and stronger post-training, all tested models remain far from reliable. As modern LLMs increasingly depend on external tools rather than parametric knowledge, this shortcoming becomes even more pronounced.

That this task proves difficult is not entirely surprising. Pretraining exposes LLMs to undifferentiated, unordered text from diverse sources without metadata, meaning that any learned ability to distinguish or compartmentalize sources must rely largely on superficial stylistic cues. This challenge is compounded by the fact that LLMs rarely see multiple documents on the same subject during training (with a few exceptions; e.g., Shi et al. (2024a)), and so detecting contradictions or inconsistencies between documents requires falling back on existing parametric knowledge, which, again, is not cleanly attributed.

Our current setup has clear limitations. Most important is the fact that we use synthetic documents. While we demonstrate that the same trends hold for real data, it is still true that instruction-tuned language models are not capable of perfectly reproducing the text distribution of the various domains in our benchmark. As such, for our synthetic results, internal LLM mechanisms that depend on the

finer details of these distributions rather than the explicit URL provided with each document may not be fairly tested. Furthermore, the fact that we use synthetic factual information throughout both datasets is also unideal; while it is desirable to ensure that models cannot rely at all on parametric knowledge to answer questions correctly, models occasionally suspected during our testing that the information in question is fictional. Though it is still reasonable to expect models to follow instructions and discard untrustworthy source URLs anyway, and though there is no guarantee that they would not react the same way to real information gathered after their respective training cutoffs, this is worth noting.

Learning better epistemic priors in a robust way will be a key challenge for future work. One promising direction is conditional pretraining: prior work has shown the potential of incorporating document-level metadata such as domains or unique identifiers (Keskar et al., 2019; Khalifa et al., 2024; Gao et al., 2025). Although existing efforts remain small in scale and lack modern post-training, extending them to more capable LLMs could yield skills directly relevant to our benchmark. On the benchmarking side, future extensions could tackle the harder task of discarding *outdated* information rather than merely untrustworthy sources.

ETHICS STATEMENT

This paper presents work whose goal is to advance the field of machine learning. There are many potential societal consequences of our work, none which we feel must be specifically highlighted here.

REPRODUCIBILITY STATEMENT

Our specific data generation processes are described in Section 2.1, with further details in Appendices B and C, whereas our experiments are presented in Section 3. We (anonymously) open-source all code used for SMeL Test experiments here.

LLM USAGE

As we describe in the main paper, we used GPT-4o in our experimental pipeline to generate and/or manipulate the documents in our test sets. We also used GPT-5 to proofread and edit our (hand-written) manuscript. All edits were validated by the authors.

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

## A   CODE

All code used to run experiments is available here.

## B   DATA GENERATION (SYNTHETIC)

As described in Section 2.1, we generate synthetic SMeL Test documents about three topics: government agencies, "true crime" incidents, and natural disasters. For *ignoring dubious sources* and *resolving contradictions*, we also generate specific facts associated with each document, drawn uniformly at random from the following sets of fact types:

- Government agencies
    - Budget: Random value between $1 billion and $200 billion.
    - Employees: Number of employees. Randomly chosen somewhere between 1000 and 25000.
    - Offices: Number of office locations. Randomly chosen between 10 and 400.
    - Citizens served: Number of citizens directly served by the agency. Randomly chosen between 1 and 60 million.
    - Laws: Number of laws that govern the activities of the agency. Randomly chosen between 10 and 70.
- Crime
    - Witnesses: Number of witnesses. Randomly chosen between 2 and "more than 100".
    - Victims: Number of victims. Chosen uniformly at random between 1 and 5.
    - Days until discovery: Number of days until the crime was discovered. Randomly chosen between 2 and 7.
    - GoFundMe: Amount raised by the family of the victim(s) on GoFundMe. Randomly chosen between 5e4 and 2.5e5.
    - Perpetrators: Number of perpetrators. Randomly chosen between 1 and 4.

- Disaster

  - Deaths: Number of deaths. Randomly chosen between 10 and 1000.
  - Damages: Amount of damages, in billions of dollars. Chosen randomly between 1 and 40.
  - Donations: Donations to victims, in millions of dollars. Chosen randomly between 10 and 90.
  - Advance warning: How early the disaster was forecasted. Chosen randomly between 2 and 7 days.
  - Time to rebuild: Number of years it is expected it will take to repair the damage. Chosen randomly between 2 and 10.

Once a disaster type is selected, a fact is sampled and the passage is generated conditioned on both.

For Task 1 (ignoring dubious sources), we generate $n = 200$ entities per topic (600 total) and, for each, sample a numerical fact type from a set of five per topic (e.g., for government agencies: budget, number of employees, number of offices). Full lists appear in Appendix Section B. A concrete fact is then generated conditioned on the entity and fact type, followed by a "seed" document (to enforce consistency), and finally a full document conditioned on the fact, entity, seed, domain, and sampled style guides. Each document is thus paired with a unique, objective factual question. We use the following descriptions for each domain in the prompt:

- Encyclopedia Britannica: "Encyclopedia Britannica"

- The New York Times: "The New York Times"

- Wikipedia: "Wikipedia"

- Reddit: "Reddit"

- 4chan: "a 4chan greentext with an irreverent punchline"

- fanfiction.net: "a mediocre, semi-fictional short story"

- "Unknown": "an unhinged, rambling, conspiratorial manifesto"

The "4chan" description is worth noting; we wanted to steer generations toward the least cite-able parts of the site, allowing it to be used as an "objectively untrustworthy" source in Task 1.

For Task 2 (resolving contradictions), we use the same synthetic documents as in *ignoring dubious sources*. For each fact-document pair, we generate a perturbed version differing only in the numerical value associated with the fact. For real data, we use the news article pairs in Section 2.2, supplemented with slightly contradictory facts centered on fact types (Appendix Section C).

For Task 3 (active filtering), we again use entities from *ignoring dubious sources*, generating one unconstrained fact per domain and a corresponding document written in that domain's style, ensuring that each domain provides distinct information. We generate facts for all domains simultaneously to avoid accidental overlap.

## C  DATA GENERATION (REAL)

As described in Section 2.2, we construct controlled contradictions within real news articles by generating non-political factual statements for each article pair. We first sample a fact type—either *Shirt Color* or *Watch*—uniformly at random. We then assign two distinct values for that fact type by randomly selecting from the following predefined sets, ensuring that no value is repeated within the same pair.:

- Shirt Color: ("red", "blue", "yellow", "orange", "pink", "green", "purple").

- Watch: ("Swatch", "Rolex", "Cartier","Omega", "Patek Philippe", "Audemars Piguet", "Seiko", "Tissot", "Breitling").

## D ARTICLE MATCHING

1. Randomly sample 5,000 *potentially trustworthy* articles in increments of 500 without repetition.

2. For each sampled fake article, identify all *trustworthy* articles whose publication date is within a $\pm 5$-day window.

3. Compute textual similarity:
   - Use TF-IDF vectorization on the `text` field with `max_features=1000`.
   - Fit the TF-IDF vectorizer once on the combined corpus of all *trustworthy* articles and the sampled *potentially trustworthy* articles to prevent repeated re-fitting.
   - Transform all *trustworthy* article texts in advance and cache their TF-IDF vectors for reuse.

4. For each date-matched article pair, transform the *potentially trustworthy* article's text using the pre-fitted TF-IDF vectorizer, and calculate the cosine similarity between the *potentially trustworthy* vector and each matched *trustworthy* article vector.

5. Retain article pairs where cosine similarity is $\geq 0.7$.

## E ADDITIONAL EXPERIMENTS

### E.1 RESOLVING CONTRADICTIONS: DOES SOURCE ORDER MATTER?

During the *resolving contradictions* subtask, models are asked to answer a question with multiple competing answers in context. In our testing (during which sources were shuffled uniformly at random), no model consistently trusts the correct source. How much of this inaccuracy can be explained by the *order* of sources in context? Do models systematically trust the dubious source more if it appears first or last? To investigate, we compute the difference in model accuracy between examples where the trustworthy source happens to appear first and those where the untrustworthy one does in Table 3.

We find that some models are much more sensitive to source ordering than others. While Gemma models and o4-mini are usually invariant, Llama models systematically trust earlier sources more, and by a wide margin. By contrast, GPT-4o often trusts the last source significantly more. Nevertheless, even for these models, empirical error rates for both orderings are still nonzero in all cases; positional bias does not account for all SMeL Test mistakes.

### E.2 RESOLVING CONTRADICTIONS: THE EFFECT OF PROMPT AND EXPLICIT SOURCE URL

To test prompt dependence, we remove all instructions warning about source reliability ("No warning" in Table 1) while leaving article text and metadata intact. Performance declines substantially, indicating that models generally do not avoid untrustworthy sources unless explicitly directed, highlighting the importance of prompt design. To test source dependence, we replace all references to the original publication (both metadata and in-text) with placeholders (e.g., "Source1"), forcing models to rely solely on article content ("No URL" in the figure/table). Under this condition, performance deteriorates markedly across all models, demonstrating a strong reliance on explicit domain names rather than intrinsic article content when judging trustworthiness.

## F EXPERIMENTAL DETAILS

### F.1 TECHNICAL DETAILS

All local experiments were run on a pair of 80GB NVIDIA H100 GPUs.

Answers to questions were sampled greedily. Passages were sampled with temperature 0.7.

### F.2 MODEL VERSIONS

We used the following versions of the API models listed in the paper:

Table 3: **Certain models are (spuriously) sensitive to source ordering**. Differences in accuracies (as percentages) on the *resolving contradictions* subtask between cases where the trustworthy source appears before the untrustworthy source and cases where it doesn't. 95% Wald confidence intervals are given for each difference. Intervals not containing zero are highlighted in red.
EB = Encyclopedia Britannica, NYT = New York Times, Wiki = Wikipedia

| Source pair | | Model | | | |
|---|---|---|---|---|---|
| Reliable | Unreliable | Gemma 3 27B | Llama 3.3 70B | GPT-4o | o3-mini |
| EB | Reddit | [-16.1, -1.0] | [9.0, 24.4] | [-21.7, -7.4] | [-3.6, 1.8] |
| NYT | | [-4.7, 11.0] | [8.2, 24.0] | [-24.1, -9.2] | [-1.2, 5.7] |
| Wiki | | [-8.7, 5.6] | [12.0, 26.7] | [-29.0, -15.4] | [-2.1, 4.0] |
| EB | 4chan | [-5.6, 5.4] | [9.1, 21.5] | [-11.9, -2.8] | [-2.3, 1.3] |
| NYT | | [-7.6, 4.4] | [5.2, 18.9] | [-9.2, 1.3] | [-2.9, 3.3] |
| Wiki | | [-7.4, 3.5] | [4.2, 16.2] | [-10.1, -0.4] | [-3.5, 2.1] |
| EB | Fan fiction | [-8.1, 5.4] | [6.3, 21.1] | [-25.8, -12.4] | [-3.4, 2.9] |
| NYT | | [-1.9, 13.0] | [5.9, 21.3] | [-19.7, -5.6] | [0.6, 9.0] |
| Wiki | | [-4.4, 8.5] | [7.4, 21.9] | [-22.7, -8.9] | [-6.4, -0.1] |
| EB | Unknown | [-10.1, 1.8] | [-0.4, 14.5] | [-6.3, 3.7] | [0.2, 5.5] |
| NYT | | [-16.0, -1.5] | [5.0, 20.7] | [-7.9, 4.0] | [-4.9, 3.5] |
| Wiki | | [-7.9, 3.9] | [5.3, 19.7] | [-9.0, 2.3] | [-2.5, 3.8] |

- OpenAI GPT-5: `gpt-5-2025-08-07`
- OpenAI GPT-4o: `chatgpt-4o-latest` (for generation) and `gpt-4o-2024-08-06` (for evaluation)
- OpenAI o3-mini: `o3-mini-2025-01-31`
- OpenAI o4-mini: `o4-mini-2025-04-16`
- OpenAI o3: `o3-2025-04-16`
- Gemini 2.5 Pro: `gemini-2.5-pro-preview-03-25`
- Claude 4.5 Sonnet: `claude-sonnet-4-5-20250929`
- Claude 3.7 Sonnet: `claude-3-7-sonnet-20250219`

F.3 PROMPTS

Here, we report the main prompts used for the three subtasks in the benchmark.

**Ignoring dubious sources:** `You are an assistant that answers questions correctly. Write nothing but the answer to each question, or "I don't know" if you don't know the answer. Write exactly one answer per question (do not write "or"). You may (but do not have to) consult the provided context. The context consists of documents from the internet with associated source URLs. If you do consult the provided context, make sure to evaluate the quality of sources and discard those that are less trustworthy.`

**Resolving contradictions:** `You are an assistant that answers questions correctly. Write nothing but the answer to each question, or "I don't know" if you don't know the answer. Write exactly one answer per question (do not write "or"). You may (but do not have to) consult the provided context. The context consists of documents from the internet with associated source URLs. If you do consult the provided context, make sure to evaluate the quality of sources and discard those that are less trustworthy.`

**Active filtering:** `You are an assistant that writes informative summaries. Given the name of a {U.S. federal government agency/famous crime/natural disaster}, write a factual and informative passage about the disaster. You may (but do not have to) consult the provided context. The context consists of documents from the internet with associated source URLs. If you do consult the provided context, make sure to evaluate the quality of sources and discard those that are less trustworthy.`

Context was then formatted as follows:

`Context document i source: {url}`
`Context document i: {document}`

For more prompts, including those used to generate the synthetic documents, see our GitHub repository.

# G  ADDITIONAL DATA

In this section, we provide additional figures and numerical data corresponding to the graphs in the main paper, as well as data for models not included above.

Table 4: **Ignoring dubious sources: No model is capable of ignoring unreliable information in context**. Hallucination rates (%, ↓) for LLMs answering straightforward factual questions ($N = 600$) for which a low-quality source in context provides the answer. A hallucination occurs when the LLM fails to abstain despite being explicitly told to ignore the unreliable source. 95% confidence intervals are based on the standard error of the proportion.

| Source | Model | | | |
|---|---|---|---|---|
| | GPT-4o | o3 | o3-mini | o4-mini |
| 4chan | $99.5 \pm 0.6$ | $99.2 \pm 0.7$ | $98.2 \pm 1.1$ | $95.8 \pm 1.6$ |
| Fan fiction | $99.8 \pm 0.4$ | $99.8 \pm 0.4$ | $98.3 \pm 1.0$ | $96.7 \pm 1.4$ |
| "Unknown" | $99.3 \pm 0.7$ | $100.0 \pm 0.0$ | $99.7 \pm 0.4$ | $99.7 \pm 0.4$ |
| | GPT-5 | Claude 3.7 Sonnet | Claude 4.5 Sonnet | Gemini 2.5 Pro |
| 4chan | $51.5 \pm 4.0$ | $97.3 \pm 1.3$ | $\mathbf{16.7 \pm 3.0}$ | $37.3 \pm 7.7$ |
| Fan fiction | $83.2 \pm 3.0$ | $99.8 \pm 0.4$ | $\mathbf{56.8 \pm 4.0}$ | $71.3 \pm 7.2$ |
| "Unknown" | $74.3 \pm 3.5$ | $83.2 \pm 3.0$ | $\mathbf{18.9 \pm 3.1}$ | $46.7 \pm 8.0$ |
| | Llama 3.1 8B | Llama 3.3 70B | Gemma 3 4B | Gemma 3 27B |
| 4chan | $89.3 \pm 2.5$ | $90.5 \pm 2.3$ | $99.3 \pm 0.7$ | $100.0 \pm 0.0$ |
| Fan fiction | $91.8 \pm 2.2$ | $91.2 \pm 2.3$ | $99.2 \pm 0.7$ | $100.0 \pm 0.0$ |
| "Unknown" | $96.2 \pm 1.5$ | $96.2 \pm 1.5$ | $100.0 \pm 0.0$ | $100.0 \pm 0.0$ |

Table 5: **Resolving contradictions (synthetic data, part 1): No model consistently prioritizes reliable sources over unreliable ones when the two conflict, but reasoning models do disproportionately well.** Hallucination rates (%, ↓) for LLMs answering straightforward factual questions ($N = 600$) based on two directly contradictory sources in context. A hallucination occurs when the model does not produce the correct answer despite being explicitly told to ignore the unreliable source. 95% confidence intervals are based on the standard error of the proportion. For part 2, see Table 6.

EB = Encyclopedia Britannica, NYT = New York Times, Wiki = Wikipedia

| Source pair | | Model | | | |
|---|---|---|---|---|---|
| Reliable | Unreliable | GPT-4o | o3 | o3-mini | o4-mini |
| EB | Reddit | $27.7 \pm 3.6$ | $1.5 \pm 1.0$ | $2.8 \pm 1.3$ | $\mathbf{1.5 \pm 1.0}$ |
| NYT | | $33.8 \pm 3.8$ | $5.0 \pm 1.7$ | $4.8 \pm 1.7$ | $6.3 \pm 1.9$ |
| Wiki | | $26.3 \pm 3.5$ | $3.5 \pm 1.5$ | $3.8 \pm 1.5$ | $\mathbf{3.0 \pm 1.4}$ |
| | | GPT-5 | Claude 3.7 Sonnet | Claude 4.5 Sonnet | Gemini 2.5 Pro |
| EB | | $\mathbf{1.5 \pm 1.0}$ | $25.3 \pm 3.5$ | $56.3 \pm 4.0$ | $8.0 \pm 2.2$ |
| NYT | | $\mathbf{3.2 \pm 1.4}$ | $34.0 \pm 3.8$ | $63.7 \pm 3.8$ | $12.7 \pm 2.7$ |
| Wiki | | $27.7 \pm 3.6$ | $30.0 \pm 3.7$ | $64.5 \pm 3.8$ | $29.3 \pm 3.6$ |
| | | Llama 3.1 8B | Llama 3.3 70B | Gemma 3 4B | Gemma 3 27B |
| EB | | $37.7 \pm 3.9$ | $40.7 \pm 3.9$ | $36.0 \pm 3.8$ | $32.3 \pm 3.7$ |
| NYT | | $45.2 \pm 4.0$ | $45.8 \pm 4.0$ | $48.2 \pm 4.0$ | $40.3 \pm 3.9$ |
| Wiki | | $34.5 \pm 3.8$ | $33.5 \pm 3.8$ | $37.3 \pm 3.9$ | $27.2 \pm 3.6$ |
| Reliable | Unreliable | GPT-4o | o3 | o3-mini | o4-mini |
| EB | 4chan | $10.3 \pm 2.4$ | $1.3 \pm 0.9$ | $1.3 \pm 0.9$ | $1.3 \pm 0.9$ |
| NYT | | $13.0 \pm 2.7$ | $3.0 \pm 1.4$ | $3.8 \pm 1.5$ | $4.2 \pm 1.6$ |
| Wiki | | $10.2 \pm 2.4$ | $2.5 \pm 1.2$ | $3.2 \pm 1.4$ | $\mathbf{2.3 \pm 1.2}$ |
| | | GPT-5 | Claude 3.7 Sonnet | Claude 4.5 Sonnet | Gemini 2.5 Pro |
| EB | | $\mathbf{0.7 \pm 0.7}$ | $7.3 \pm 2.1$ | $19.8 \pm 3.2$ | $2.7 \pm 1.3$ |
| NYT | | $\mathbf{2.0 \pm 1.1}$ | $20.0 \pm 3.2$ | $27.2 \pm 3.6$ | $6.7 \pm 2.0$ |
| Wiki | | $15.8 \pm 2.9$ | $13.3 \pm 2.7$ | $19.8 \pm 3.2$ | $5.3 \pm 1.8$ |
| | | Llama 3.1 8B | Llama 3.3 70B | Gemma 3 4B | Gemma 3 27B |
| EB | | $19.7 \pm 3.2$ | $18.3 \pm 3.1$ | $14.7 \pm 2.8$ | $13.7 \pm 2.8$ |
| NYT | | $25.3 \pm 3.5$ | $24.2 \pm 3.4$ | $25.2 \pm 3.5$ | $17.0 \pm 3.0$ |
| Wiki | | $21.0 \pm 3.3$ | $18.2 \pm 3.1$ | $16.7 \pm 3.0$ | $13.2 \pm 2.7$ |

Table 6: **Resolving contradictions (synthetic data, part 2): No model consistently prioritizes reliable sources over unreliable ones when the two conflict, but reasoning models do disproportionately well.** Hallucination rates (%, ↓) for LLMs answering straightforward factual questions ($N = 600$) based on two directly contradictory sources in context. A hallucination occurs when the model does not produce the correct answer despite being explicitly told to ignore the unreliable source. 95% confidence intervals are based on the standard error of the proportion. For part 1, see Table 5.

EB = Encyclopedia Britannica, NYT = New York Times, Wiki = Wikipedia

| Source pair | | Model | | | |
| --- | --- | --- | --- | --- | --- |
| Reliable | Unreliable | GPT-4o | o3 | o3-mini | o4-mini |
| EB | Fan fiction | $24.3 \pm 3.4$ | $2.8 \pm 1.3$ | $4.2 \pm 1.6$ | $2.3 \pm 1.2$ |
| NYT | | $28.3 \pm 3.6$ | $8.0 \pm 2.2$ | $7.7 \pm 2.1$ | $7.2 \pm 2.1$ |
| Wiki | | $26.3 \pm 3.5$ | $3.3 \pm 1.4$ | $4.0 \pm 1.6$ | $\mathbf{2.3 \pm 1.2}$ |
| | | GPT-5 | Claude 3.7 Sonnet | Claude 4.5 Sonnet | Gemini 2.5 Pro |
| EB | | $\mathbf{1.7 \pm 1.0}$ | $14.0 \pm 2.8$ | $53.3 \pm 4.0$ | $6.7 \pm 2.0$ |
| NYT | | $\mathbf{4.5 \pm 1.7}$ | $28.0 \pm 3.6$ | $57.8 \pm 4.0$ | $9.3 \pm 2.3$ |
| Wiki | | $26.2 \pm 3.5$ | $24.7 \pm 3.5$ | $55.5 \pm 4.0$ | $16.0 \pm 2.9$ |
| | | Llama 3.1 8B | Llama 3.3 70B | Gemma 3 4B | Gemma 3 27B |
| EB | | $25.5 \pm 3.5$ | $33.0 \pm 3.8$ | $24.2 \pm 3.4$ | $23.0 \pm 3.4$ |
| NYT | | $32.0 \pm 3.7$ | $37.8 \pm 3.9$ | $31.3 \pm 3.7$ | $31.2 \pm 3.7$ |
| Wiki | | $24.0 \pm 3.4$ | $30.3 \pm 3.7$ | $22.7 \pm 3.4$ | $20.5 \pm 3.2$ |
| Reliable | Unreliable | GPT-4o | o3 | o3-mini | o4-mini |
| EB | Unknown | $11.2 \pm 2.5$ | $1.8 \pm 1.1$ | $3.2 \pm 1.4$ | $2.7 \pm 1.3$ |
| NYT | | $16.8 \pm 3.0$ | $4.7 \pm 1.7$ | $7.3 \pm 2.1$ | $5.7 \pm 1.9$ |
| Wiki | | $15.2 \pm 2.9$ | $27.7 \pm 3.6$ | $4.0 \pm 1.6$ | $\mathbf{3.2 \pm 1.4}$ |
| | | GPT-5 | Claude 3.7 Sonnet | Claude 4.5 Sonnet | Gemini 2.5 Pro |
| EB | | $\mathbf{1.7 \pm 1.0}$ | $14.0 \pm 2.8$ | $25.5 \pm 3.5$ | $2.7 \pm 1.3$ |
| NYT | | $\mathbf{2.8 \pm 1.3}$ | $14.7 \pm 2.8$ | $27.8 \pm 3.6$ | $6.7 \pm 2.0$ |
| Wiki | | $16.7 \pm 3.0$ | $10.7 \pm 2.5$ | $31.3 \pm 3.7$ | $6.0 \pm 1.9$ |
| | | Llama 3.1 8B | Llama 3.3 70B | Gemma 3 4B | Gemma 3 27B |
| EB | | $30.7 \pm 3.7$ | $32.5 \pm 3.7$ | $31.7 \pm 3.7$ | $15.7 \pm 2.9$ |
| NYT | | $41.0 \pm 3.9$ | $43.0 \pm 4.0$ | $41.7 \pm 3.9$ | $28.8 \pm 3.6$ |
| Wiki | | $27.2 \pm 3.6$ | $29.7 \pm 3.7$ | $30.0 \pm 3.7$ | $16.3 \pm 3.0$ |

Table 7: **Active filtering (part 1): No LLM successfully insulates its generations from untrustworthy sources in context.** Hallucination rates (%, ↓) for LLMs generating summaries ($N = 600$) based on two sources in context. A hallucination occurs when a grader LLM indicates that the unreliable source influenced the summary despite instructions to ignore it. 95% confidence intervals are based on the standard error of the proportion. Note that Gemini 2.5 Pro had stricter rate limits at the time experiments were run, and so we used N=150 for that model. See Table 8 for part 2. EB = Encyclopedia Britannica, NYT = New York Times, Wiki = Wikipedia

| Source pair | | Model | | | |
|---|---|---|---|---|---|
| Reliable | Unreliable | GPT-4o | o3 | o3-mini | o4-mini |
| EB | Reddit | $60.2 \pm 3.9$ | $60.8 \pm 3.9$ | $61.2 \pm 3.9$ | $68.2 \pm 3.7$ |
| NYT | | $79.3 \pm 3.2$ | $86.7 \pm 2.7$ | $77.8 \pm 3.3$ | $78.8 \pm 3.3$ |
| Wiki | | $72.3 \pm 3.6$ | $70.0 \pm 3.7$ | $68.7 \pm 3.7$ | $72.8 \pm 3.6$ |
| | | GPT-5 | Claude 3.7 Sonnet | Claude 4.5 Sonnet | Gemini 2.5 Pro |
| EB | | $\mathbf{12.5 \pm 2.6}$ | $83.0 \pm 3.0$ | $86.7 \pm 2.7$ | $57.3 \pm 7.9$ |
| NYT | | $\mathbf{12.0 \pm 2.6}$ | $91.3 \pm 2.3$ | $85.0 \pm 2.9$ | $63.3 \pm 7.7$ |
| Wiki | | $\mathbf{20.2 \pm 3.2}$ | $86.7 \pm 2.7$ | $85.2 \pm 2.8$ | $67.3 \pm 7.5$ |
| | | Llama 3.1 8B | Llama 3.3 70B | Gemma 3 4B | Gemma 3 27B |
| EB | | $65.5 \pm 3.8$ | $78.5 \pm 3.3$ | $76.7 \pm 3.4$ | $88.5 \pm 2.6$ |
| NYT | | $75.2 \pm 3.5$ | $83.0 \pm 3.0$ | $80.7 \pm 3.2$ | $90.1 \pm 2.4$ |
| Wiki | | $69.3 \pm 3.7$ | $81.3 \pm 3.1$ | $80.0 \pm 3.2$ | $90.7 \pm 2.3$ |
| Reliable | Unreliable | GPT-4o | o3 | o3-mini | o4-mini |
| EB | 4chan | $19.7 \pm 3.2$ | $40.7 \pm 3.9$ | $14.0 \pm 2.8$ | $23.7 \pm 3.4$ |
| NYT | | $31.2 \pm 3.7$ | $54.3 \pm 4.0$ | $20.8 \pm 3.2$ | $29.8 \pm 3.7$ |
| Wiki | | $27.5 \pm 3.6$ | $50.3 \pm 4.0$ | $16.2 \pm 2.9$ | $29.7 \pm 3.7$ |
| | | GPT-5 | Claude 3.7 Sonnet | Claude 4.5 Sonnet | Gemini 2.5 Pro |
| EB | | $\mathbf{5.7 \pm 1.9}$ | $57.0 \pm 4.0$ | $37.3 \pm 3.9$ | $6.7 \pm 4.0$ |
| NYT | | $5.2 \pm 1.8$ | $66.0 \pm 3.8$ | $46.0 \pm 4.0$ | $\mathbf{4.7 \pm 3.4}$ |
| Wiki | | $\mathbf{7.8 \pm 2.2}$ | $68.7 \pm 3.7$ | $47.3 \pm 4.0$ | $10.7 \pm 4.9$ |
| | | Llama 3.1 8B | Llama 3.3 70B | Gemma 3 4B | Gemma 3 27B |
| EB | | $30.5 \pm 3.7$ | $45.7 \pm 4.0$ | $46.2 \pm 4.0$ | $57.6 \pm 4.0$ |
| NYT | | $39.5 \pm 3.9$ | $49.7 \pm 4.0$ | $50.5 \pm 4.0$ | $66.7 \pm 3.8$ |
| Wiki | | $35.5 \pm 3.8$ | $47.2 \pm 4.0$ | $52.7 \pm 4.0$ | $60.7 \pm 3.9$ |

Table 8: **Active filtering (part 2): No LLM successfully insulates its generations from untrustworthy sources in context.** Hallucination rates (%, ↓) for LLMs generating summaries ($N = 600$) based on two sources in context. ==A hallucination occurs when a grader LLM indicates that the unreliable source influenced the summary despite instructions to ignore it.== 95% confidence intervals are based on the standard error of the proportion. Note that Gemini 2.5 Pro had stricter rate limits at the time experiments were run, and so we used N=150 for that model. See Table 7 for part 1. EB = Encyclopedia Britannica, NYT = New York Times, Wiki = Wikipedia

| Source pair | | Model | | | |
|---|---|---|---|---|---|
| Reliable | Unreliable | GPT-4o | o3 | o3-mini | o4-mini |
| EB | Fan fiction | $29.5 \pm 3.6$ | $48.7 \pm 4.0$ | $26.2 \pm 3.5$ | $33.5 \pm 3.8$ |
| NYT | | $45.7 \pm 4.0$ | $56.2 \pm 4.0$ | $43.3 \pm 4.0$ | $41.5 \pm 3.9$ |
| Wiki | | $41.8 \pm 3.9$ | $52.5 \pm 4.0$ | $36.7 \pm 3.9$ | $38.7 \pm 3.9$ |
| | | GPT-5 | Claude 3.7 Sonnet | Claude 4.5 Sonnet | Gemini 2.5 Pro |
| EB | | $8.0 \pm 2.2$ | $79.0 \pm 3.3$ | $63.7 \pm 3.8$ | $\mathbf{6.7 \pm 4.0}$ |
| NYT | | $\mathbf{6.5 \pm 2.0}$ | $84.7 \pm 2.9$ | $63.8 \pm 3.8$ | $10.0 \pm 4.8$ |
| Wiki | | $\mathbf{10.2 \pm 2.4}$ | $77.3 \pm 3.4$ | $67.7 \pm 3.7$ | $24.7 \pm 6.9$ |
| | | Llama 3.1 8B | Llama 3.3 70B | Gemma 3 4B | Gemma 3 27B |
| EB | | $35.2 \pm 3.8$ | $52.3 \pm 4.0$ | $54.5 \pm 4.0$ | $62.3 \pm 3.9$ |
| NYT | | $42.0 \pm 3.9$ | $56.7 \pm 4.0$ | $58.3 \pm 3.9$ | $69.3 \pm 3.7$ |
| Wiki | | $40.0 \pm 3.9$ | $54.0 \pm 4.0$ | $56.2 \pm 4.0$ | $63.5 \pm 3.9$ |
| Reliable | Unreliable | GPT-4o | o3 | o3-mini | o4-mini |
| EB | Unknown | $20.5 \pm 3.2$ | $42.2 \pm 4.0$ | $17.0 \pm 3.0$ | $26.8 \pm 3.5$ |
| NYT | | $33.3 \pm 3.8$ | $42.3 \pm 4.0$ | $26.7 \pm 3.5$ | $31.3 \pm 3.7$ |
| Wiki | | $29.0 \pm 3.6$ | $43.5 \pm 4.0$ | $23.7 \pm 3.4$ | $26.0 \pm 3.5$ |
| | | GPT-5 | Claude 3.7 Sonnet | Claude 4.5 Sonnet | Gemini 2.5 Pro |
| EB | | $\mathbf{4.0 \pm 1.6}$ | $32.0 \pm 3.7$ | $27.5 \pm 3.6$ | $8.0 \pm 4.3$ |
| NYT | | $\mathbf{4.8 \pm 1.7}$ | $37.1 \pm 3.9$ | $32.0 \pm 3.7$ | $6.7 \pm 4.0$ |
| Wiki | | $\mathbf{8.0 \pm 2.2}$ | $46.7 \pm 4.0$ | $27.5 \pm 3.6$ | $12.0 \pm 5.2$ |
| | | Llama 3.1 8B | Llama 3.3 70B | Gemma 3 4B | Gemma 3 27B |
| EB | | $38.2 \pm 3.9$ | $40.2 \pm 3.9$ | $64.2 \pm 3.8$ | $52.0 \pm 4.0$ |
| NYT | | $47.5 \pm 4.0$ | $52.8 \pm 4.0$ | $72.2 \pm 3.6$ | $64.5 \pm 3.8$ |
| Wiki | | $40.8 \pm 3.9$ | $48.5 \pm 4.0$ | $67.1 \pm 3.8$ | $58.2 \pm 3.9$ |

