# OpenReview forum: "The SMeL Test: A simple benchmark for media literacy in language models"
_ICLR.cc/2026/Conference — Submitted to ICLR 2026_

### Official Review · Reviewer_De3v · 2025-10-27

**Soundness:** 2
**Presentation:** 1
**Contribution:** 2
**Rating:** 2
**Confidence:** 3

**Summary:**

This paper introduces SMeL Test benchmark (Synthetic Media Literacy Test) to evaluate the ability of LLMs to identify and filter out untrustworthy information in context. The benchmark consists of three tasks that require awareness of source quality, i.e. 1). question answering given an untrustworthy source 2). question answering given a pair of reliable source and unreliable source with contradictory information 3). generate a summary that filters out untrustworthy information. To obtain input documents with different levels of trustworthiness, they use GPT model to generate synthetic documents that mimic the styles of different websites. For each task, they mimic real-world RAG setting by providing a mixture of these synthetic documents and some randomly sampled irrelevant documents as input context. They evaluate both open LLMs and API-based models on SMeL benchmark and measure the model performance by hallucination rate (i.e. how often the untrustworthy information is used by LLM). Their experiments reveal that even though LLMs can explicitly recognize untrustworthy sources, they still use untrustworthy information to perform the tasks. Increasing model sizes does not bring significant improvement in their benchmark. Reasoning models generally outperform non-reasoning ones.

**Strengths:**

- This paper aims to address a critical challenge in RAG system by assessing the ability of LLMs to identify and filter out untrustworthy information when the input context contains a mixture of information sources
- They propose a new benchmark called SMeL Test based on synthetic data to measure how often the LLMs are affected by untrustworthy information in question answering and summarisation
- They conduct experiments with diverse state-of-the-art API models on SMeL Test benchmark and perform qualitative analysis

**Weaknesses:**

- All the documents in SMeL Test are generated by GPT-4o and contain fictional information. The trustworthiness of information is not clearly defined since there is no ground truth.
- Model performance can be highly affected by the prompt instructions and positional bias, which may compromise the validity of the conclusions.
- The writing style is not sufficiently academic. Some word choices and sentences are somewhat informal, e.g. ”deep research products consistently err”, “we say a hallucination occurs when…”, "reasoning models do better", “(Mostly) Real data”

**Questions:**

- Table 1 could be improved by using clearer and more descriptive row names, and the caption or paragraph should explain more details about how these numbers are calculated
- In Figure 4, the evaluation metric of Gemini 2.5 Pro is computed on a different number of examples. The comparison among different models is not fair enough.

---

> ### Author Response · Authors · 2025-11-15
>
> We thank the reviewer for their time and respond to their individual concerns below:
>
> > All the documents in SMeL Test are generated by GPT-4o and contain fictional information. The trustworthiness of information is not clearly defined since there is no ground truth.
>
> While we generate synthetic documents for SMeL, we also try to provide evidence in the paper that the synthetic nature of the test documents does not have a significant effect on the results: first, our parallel results on a dataset of real documents show similar trends to the synthetic results (Table 1). Secondly, we note that models can consistently identify which source is more reliable when asked directly (they never claim something like “both sources are equally trustworthy because they are synthetic”),
>
> Furthermore, in terms of our synthetic document generation, while the documents we use contain fictional information, we use real domains. The “ground truth” in this case is the real fact that Encyclopedia Britannica is a more trustworthy source than fictional short stories, and we feel that this distinction is clearly defined.
>
> > Model performance can be highly affected by the prompt instructions and positional bias, which may compromise the validity of the conclusions.
>
> The same could be said of any LLM benchmark. In our case, in addition to experimenting with variations of our main prompts (e.g. table 1), we took care to word our main prompts in a fair way. We even go as far as giving the models explicit instructions to ignore dubious sources, which may not be present in real Q/A interactions with chatbots (see Appendix F for the main task prompts). We argue that a failure under these conditions is worth noting and, indeed, testing.
>
> As for positional bias, we include experiments to test the effect of position on results in Table 2 of the appendix. While some models are sensitive to source order, others are not, and the effect is not large enough to explain the failures of models on the SMeL tasks. Even if there were some order that yielded much stronger results, we would still argue that SMeL tasks should allow the models to succeed regardless of source ordering.
>
> > The writing style is not sufficiently academic. Some word choices and sentences are somewhat informal, e.g. ”deep research products consistently err”, “we say a hallucination occurs when…”, "reasoning models do better", “(Mostly) Real data”
>
> We regret that the reviewer found the style objectionable. While we are cognizant of the difficulty of litigating this over OpenReview, we are not sure that we agree with the reviewer’s characterization overall. “err” is probably overly formal, and  “we say” is a common phrase in technical definitions. Of course, “do better” could definitely be phrased more precisely, and “(Mostly) Real” is a little tongue-in-cheek—we are more than happy to work with you on this. In the most recent version of the manuscript, we have removed these particular phrases in response to your feedback and reworded all of the similar ones we could find throughout. Changes are highlighted in yellow.
>
> We do want to argue that, since ICLR does not mandate a specific writing style, this criticism would be more salient if the prose were so informal that it significantly affected the clarity of the scientific contributions in some way. If this is the case, please let us know.
>
> For fun, in the spirit of ICLR’s LLM feedback pilot last year, we thought we would ask GPT-5 to evaluate the writing style of the manuscript. See the (anonymized) chat https://chatgpt.com/share/6915365d-f898-8003-93d5-3e05f0abdaa8 here. The model is, as usual, very sycophantic, and its feedback should be taken with more than a grain of salt, but its main objection was that the prose was *too* formal: “Some phrases (e.g., “we posit three disjoint categories of sources”) are slightly formal or dated; minor rephrasing could enhance accessibility.”
>
> > Table 1 could be improved by using clearer and more descriptive row names, and the caption or paragraph should explain more details about how these numbers are calculated
>
> We have updated both the row names and also the caption of this table (now Table 2 in the most recent version of the manuscript) to make it more clear what is being shown.
>
> > In Figure 4, the evaluation metric of Gemini 2.5 Pro is computed on a different number of examples. The comparison among different models is not fair enough.
>
> Thank you for pointing this out; those results were gathered shortly after Gemini’s release, when rate limits were much stricter. Between then and the submission deadline, we decided that the additional expense of re-running those experiments might not add much to the paper, especially given that we provide confidence intervals. For consistency, however, we are currently running a full set of experiments with Gemini 2.5 Pro. We will update the manuscript as soon as they're ready.

---

> > ### Author Response · Authors · 2025-11-15
> >
> > We hope we have addressed most of the reviewer’s concerns and that you would find it fitting to raise your score. If not, please let us know how we can work together to improve the manuscript. We would be happy to make any additional necessary changes.

---

### Official Review · Reviewer_aokR · 2025-10-31

**Soundness:** 3
**Presentation:** 4
**Contribution:** 3
**Rating:** 8
**Confidence:** 4

**Summary:**

This paper proposes a new benchmark called the SMeL test, exploring whether models can recognize content from untrustworthy sources when synthesizing information. They do this by creating a series of RAG-like evals where the models receive documents and need to answer a question based on them. The results show that models do poorly at recognizing untrustworthy sources and abstaining to incorporate it into their answers. They also show some interesting case studies with reasoning vs non-reasoning models.

**Strengths:**

- The proposed topic is important and timely
- The paper is well-written and the benchmark seems relatively well-designed
- A plethora of models are evaluated and there are numerous ablations

**Weaknesses:**

I think the paper is strong but these are the minor weakness I see.

1: Some of the tasks are relatively arbitrary (e.g. Ignoring dubious sources requires the model to ignore the sources without really asking the model to do so). In many cases, I expect the model will need to be clearly told to ignore those sources. However, other sections of the eval does have this and I think this is more compelling.

2. I think as a general problem for this area of research is that whether some sources are dubious is very contentious these days and I think it is hard for model providers to be able to design for this. However, the authors choose good domains to test this (no one should argue with fan fiction being excluded).

**Questions:**

o3 is one of the strongest models for searching and finding information - I wonder if these rates are relatively low because the model has been trained to be stronger at RAG. Do you think these are conflicting or could be resolved with better training?

---

> ### Author Response · Authors · 2025-11-15
>
> We are glad that the reviewer agrees on the importance of this problem and are grateful for their strong support. We respond to their individual concerns below:
>
> > Some of the tasks are relatively arbitrary (e.g. Ignoring dubious sources requires the model to ignore the sources without really asking the model to do so)
>
> This is actually not true. The prompt for Task 1 (as well as the other tasks), includes a specific instruction to ignore dubious sources: “If you do consult the provided context, make sure to evaluate the quality of sources and discard those that are less trustworthy.” Full prompts are listed in Appendix F.
>
> > I think as a general problem for this area of research is that whether some sources are dubious is very contentious these days and I think it is hard for model providers to be able to design for this. However, the authors choose good domains to test this (no one should argue with fan fiction being excluded).
>
> We completely agree; this is a subtle problem and the right answer will depend on the application, target audience, and so on. This is why we only run Task 1 on the “objectively untrustworthy” sources, since it would be unreasonable otherwise to expect a model to categorically exclude the sources in question.
>
> > o3 is one of the strongest models for searching and finding information - I wonder if these rates are relatively low because the model has been trained to be stronger at RAG. Do you think these are conflicting or could be resolved with better training?
>
> Yes, RAG-specific training probably helps significantly. There is some evidence in our results that just scaling compute in a general way might not translate into improvements in SMeL score (e.g. smaller Gemmas are just as good as the bigger ones, and “-mini” GPT models also have a disproportionately strong showing). So one probably would need some sort of specialized instruction.

---

> > ### Comment · Reviewer_aokR · 2025-11-28
> >
> > Thanks for the updates, I now see the prompt in the appendix.  I will leave my current score as it is already high.

---

### Official Review · Reviewer_BRwb · 2025-10-31

**Soundness:** 2
**Presentation:** 2
**Contribution:** 3
**Rating:** 4
**Confidence:** 4

**Summary:**

This paper introduces Synthetic Media Literacy Test (SMeL Test), a benchmark to test how well LLMs can differentiate (and filter) between trustworthy and untrustworthy/fictional data sources when providing answers or summaries. The paper runs this benchmark on contemporary open-weight and API LLMs, and uses a case-study using real news articles to validate its efficacy.

**Benchmark data**: The benchmark consists of synthetic documents that mimic various sources with different levels of trustworthiness (Encyclopedia Britannica, NYT, Wikipedia, Reddit, 4chan, fanfiction.net, and unattributed untrustworthy documents). For a set of fictional controversial topics, the benchmark generates synthetic documents in the style of the various sources. It also includes "false-positive" documents (from the internet) that do not contain information about any of the fictional topics.

**Benchmark tasks**: The benchmark has three types of tasks:
1. Ignoring dubious sources: Given a factual question and a single document from an untrustworthy source (and unrelated false-positive documents), an LLM should abstain from providing an answer.
2. Resolving contradictions: Given a factual question and two sources with contradicting information, an LLM should rely on the more trustworthy one when answering.
3. Active filtering: An LLM is instructed to write a summary. Given two documents, one from a trustworthy and one from an untrustworthy source, the LLM should only rely on information from the trustworthy source.
In all tasks, models are explicitly instructed to ignore untrustworthy sources (and to abstain from answering for the first task), and sources are provided with their URL in-context.

**Real-world data comparison**: The paper does a case-study on real news articles to compare benchmark on synthetic and real data. The authors use the ISOT Fake News Dataset to obtain pairs of real news articles about the same topic, one from a verified trustworthy source, and one from a verified partially trustworthy source. They then create a synthetic statement for every pair of articles, and insert it with slight variations into the individual articles.

**Findings**: The paper contains many findings related to the question of how LLMs consider the trustworthiness of sources, for example
1. Larger models in the same family (e.g., Gemma 3 4B vs. 27B) do not necessarily perform better.
2. Reasoning models perform better than non-reasoning ones.
3. All models (even from different families) have a similar internal ranking of how trustworthy sources are.
4. Most models fail to abstain from answering a question if there is no reliable source.
5. Models sometimes can express which sources are more trustworthy than others, but the same models do not act accordingly.
6. For the "resolving contradictions" task, models perform worse on pairs of real articles with injected information than on pairs of synthetic documents.

**Strengths:**

**Significant topic**: The overall problem this paper studies is relevant and contemporary. Frontier models often rely on search, and the internet is becoming increasingly cluttered with untrustworthy data. Hence, understanding how well LLMs handle sources of different trustworthiness helps users understand risks and to potentially fix issues.

**Useful and rigorous datasets**: The dataset generation procedures (both the synthetic benchmark and the real-world news articles) seems to be done rigorously, and I cannot see any spurious biases. Hence, the benchmark seems sound. I also like the comparison to real news articles; the results hint that using synthetic data does not hurt the benchmark's efficacy a lot. And it seems straightforward to update/expand the synthetic dataset in the future due to its design.

**Clear goals and limitations**: The authors motivate the significance of their benchmark well, and the goals are clear. They are also transparent about the limitations of using synthetic data. The three high-level benchmark tasks (at least as described in Section 2; ignoring subsequent issues) align well with the paper's goals.

**Weaknesses:**

**Active filtering task is too limited**: The Section 2 description of the "active filtering" task mentions filtering between many sources. This is (in my opinion) the most interesting task, because it corresponds to "deep research", which is most affected by untrustworthy sources. However, later (Section 3), it becomes clear that this task only uses two documents. I think only using pairs of documents is highly restrictive and does not serve as a proxy for real-world performance of "deep research" systems; thus, currently only the first two tasks are truly insightful, and the third task is closer to the second one than to the real world.

**LLMs being benchmarked are inconsistent**: While the paper considers a broad set of LLMs, their usage throughout the paper is inconsistent; it feels almost like two separate benchmarks, with only results from one or the other shown. This makes it very hard to judge the insights from the benchmark and often conflicts with the writing. Hence, I cannot really judge whether the conclusions are valid. I believe the paper would heavily benefit from a thorough cleanup in this regard, making sure that the writing, figures, and results are consistent. Some examples:
- The experiment setup (L181-183) mentions Gemma 3, Llama 3, GPT-5, o4-mini, o3 Gemini 2.5 Pro, Claude 3.7
- The immediately referenced appendix does not mention GPT-5, so it's not clear which checkpoint was used.
- Figures 2,3,5 do not show the aforementioned Gemma 3 or Claude 3.7 models, but instead show GPT-4o.
- Table 1 mentions o3*-mini*, yet another model. This model neither shows up in Table 1 nor its "overflow" Table 10.
- The models in Figure 3 and 4 ("resolving contradictions" for synthetic and real data, respectively) are different, making a comparison impossible. Hence, I cannot easily verify the claim that "the relative performance trends among models remain consistent" on L253. However, this is the main justification for using a synthetic benchmark.
- L234-235 mention "Models in the Gemma and Llama families do not appear to improve with added size". There are two different sizes from the Gemma 3 family, but for Llama, there is only Llama 3.1 8B and Llama 3.3 70B, which are not directly comparable.
- The paper uses the newest versions of frontier models (GPT-5, Gemini 2.5 Pro), but only Claude 3.7 (not any of the 4.x models).
- GPT-4o uses the `chatgpt-4o-latest` checkpoint, but this checkpoint is generally not recommended for use by OpenAI. Instead, a regular 4o checkpoint should be used.

**Minor points**:
1. The figures (and placement) in the main matter could be improved. Currently, Figures 2-4 are all combined on a single page, far from where they are referenced in writing. Listing the figures closer to where they are referenced would make the flow easier. For example, Figures 3 and 4 could be combined into two subfigures. Those two figures would also benefit from being homogenized; currently they use different models and have different x axes, which makes the comparison between synthetic and real data hard to assess.
2. There is a non-anonymous URL on L1078-1079.

**Questions:**

1. What is the reliable source in Figure 3? Is it averaged over EB+NYT+Wiki, or only one of them?
2. Why does the real-world dataset not use untrustworthy sources (only partially trustworthy ones)? Using fully untrustworthy sources might reduce the gap in hallucination rate between synthetic and real data.
3. The paper explicitly introduces three categories of sources ("trustworthy", "potentially trustworthy", "objectively untrustworthy"). How do the 7 sources in Section 2.1 map into those categories? I could not find anything relating to that in the paper.

---

> ### Author Response · Authors · 2025-11-15
>
> We thank the reviewer for their careful reading and thorough review. We appreciate your feedback and below include answers to your questions. Note that we have uploaded a new version of the manuscript with the most important changes highlighted in yellow.
>
> > However, later (Section 3), it becomes clear that this task only uses two documents. I think only using pairs of documents is highly restrictive and does not serve as a proxy for real-world performance of "deep research" systems; thus, currently only the first two tasks are truly insightful, and the third task is closer to the second one than to the real world.
>
> We agree that task 3 in its current state is easier than the real-world retrieval tasks that inspired it. However, we’d argue that that’s exactly the point: the fact that models do not saturate even the simplest possible version of this benchmark is a surprising new finding that is worth noting. Furthermore, while the task in its current form is certainly similar to task 2, we note that task 3 performance is significantly worse across the board, indicating that the option to include information from all available sources (as opposed to just choosing one) poses a significant new challenge for the models.
> That being said, one of the advantages of the SMeL Test is that it can easily be expanded and modified. We would be happy to include multi-document versions of task 3. Please let us know if the reviewer believes this would benefit the paper.
>
> >  While the paper considers a broad set of LLMs, their usage throughout the paper is inconsistent; it feels almost like two separate benchmarks, with only results from one or the other shown. This makes it very hard to judge the insights from the benchmark and often conflicts with the writing.
>
> This is a fair criticism—we had been adding new models to the manuscript as they were released, and in retrospect we agree that it became a little disorganized. We have results for all of the models in question, and have uploaded a new version of the manuscript that addresses the issues you identify here.
>
> > L234-235 mention "Models in the Gemma and Llama families do not appear to improve with added size". There are two different sizes from the Gemma 3 family, but for Llama, there is only Llama 3.1 8B and Llama 3.3 70B, which are not directly comparable.
>
> We agree that our wording here was imprecise. Unfortunately, there is no Llama 3.3 8B model, and so we had the choice of using the older Llama 3.1 70B or the newer 3.3 version. We chose the newer model to be as fair as possible. We have updated the wording of this section accordingly (as well as having made clear that the same holds for GPT 4/5  and corresponding “mini” models).
>
> > The paper uses the newest versions of frontier models (GPT-5, Gemini 2.5 Pro), but only Claude 3.7 (not any of the 4.x models).
>
> We added Claude 3.7 to the manuscript before the release of the 4.x models. Since its performance was comparable to that of GPT models of the same age, we prioritized GPT-5 and Gemini in the newer version over additional Claude models.
> However, in response to this feedback, we are currently running experiments with Claude 4.5 Sonnet. We will follow up with a new version of the manuscript as soon as we have results.
>
> > GPT-4o uses the chatgpt-4o-latest checkpoint, but this checkpoint is generally not recommended for use by OpenAI. Instead, a regular 4o checkpoint should be used.
>
> We used `chatgpt-4o-latest` specifically to generate synthetic documents, as we found it far superior to the API checkpoint at this task. For the SMeL evaluation, we actually did use a regular 4o checkpoint (`gpt-4o-2024-08-06`), as you recommend. We have added a line to the newest version of the manuscript to reflect this.
>
> > The figures (and placement) in the main matter could be improved. Currently, Figures 2-4 are all combined on a single page, far from where they are referenced in writing. Listing the figures closer to where they are referenced would make the flow easier. For example, Figures 3 and 4 could be combined into two subfigures. Those two figures would also benefit from being homogenized; currently they use different models and have different x axes, which makes the comparison between synthetic and real data hard to assess.
>
> We have reorganized the figures as requested. Please let us know if there are any additional concerns about their placement.
> Figures 3 and 4 in the original version of the manuscript were difficult to homogenize because the real dataset evaluation uses different source pairings. We agree that the proximity of these figures was confusing, however, and have moved Figure 4 to the appendix, replacing it with Table 1. This should be easier to compare to the synthetic results.
>
> > There is a non-anonymous URL on L1078-1079.
>
> We apologize for this oversight. We have removed the URL from the most recent version of the manuscript.

---

> ### Author Response · Authors · 2025-11-15
>
> > What is the reliable source in Figure 3? Is it averaged over EB+NYT+Wiki, or only one of them?
>
> It’s an average. We have updated the caption of this and similar figures to make this more clear.
>
> > Why does the real-world dataset not use untrustworthy sources (only partially trustworthy ones)? Using fully untrustworthy sources might reduce the gap in hallucination rate between synthetic and real data.
>
> We agree that fully untrustworthy sources might help to close the gap. With the real data experiments we did not intend to make the point that the model performs categorically worse on real data; we just wanted to ground results on the synthetic evaluations, which, as we note in the paper, have their own advantages. We did not include fully untrustworthy real sources only because our problem setting is quite novel, and to our knowledge there does not exist a dataset of paired articles that matches our requirements (namely, where the pairs of articles are of radically differing quality and also on the same topic).
>
> > The paper explicitly introduces three categories of sources ("trustworthy", "potentially trustworthy", "objectively untrustworthy"). How do the 7 sources in Section 2.1 map into those categories? I could not find anything relating to that in the paper.
>
> Thank you for pointing this out. We have added a breakdown to section 2.1.
>
> Thank you again for the useful suggestions; we feel that the manuscript has been substantially improved as a result. Please let us know if there are any remaining issues and whether you would be willing to raise your score.

---

> ### Author Response · Authors · 2025-11-18
>
> We've run Claude 4.5 Sonnet. It's an interesting one: it's the new leader on Task 1 (by a wide margin), but regresses by more than 2x on Task 2 compared to Claude 3.7 Sonnet, and is in particular 2x more likely to trust fan fiction. This fits our prior observation that model size/release date do not necessarily correlate with success on our benchmark. The results are in the updated manuscript and also reproduced here for convenience (recall that lower is better):
>
> **Overall:**
>
> | Metric           | Score      |
> | ---------------- | ---------- |
> | Task 1 (average) | 30.8 ± 3.7 |
> | Task 2 (average) | 41.9 ± 3.9 |
> | Task 3 (average) | 55.8 ± 4.0 |
> | SMeL score       | 42.8 ± 4.0 |
>
> **Task 1:**
>
> | Source      |  |
> | ----------- | ----------------- |
> | 4chan       | 16.7 ± 3.0        |
> | Fan fiction | 56.8 ± 4.0        |
> | "Unknown"   | 18.9 ± 3.1        |
>
> **Task 2:**
>
> | Reliable \ Unreliable | Reddit     | 4chan      | Fan fiction | "Unknown"  |
> | --------------------- | ---------- | ---------- | ----------- | ---------- |
> | EB                    | 56.3 ± 4.0 | 19.8 ± 3.2 | 53.3 ± 4.0  | 25.5 ± 3.5 |
> | NYT                   | 63.7 ± 3.8 | 27.2 ± 3.6 | 57.8 ± 4.0  | 27.8 ± 3.6 |
> | Wiki                  | 64.5 ± 3.8 | 19.8 ± 3.2 | 55.5 ± 4.0  | 31.3 ± 3.7 |
>
> **Task 3:**
>
> | Reliable \ Unreliable | Reddit     | 4chan      | Fan fiction | "Unknown"  |
> | --------------------- | ---------- | ---------- | ----------- | ---------- |
> | EB                    | 86.7 ± 2.7 | 37.3 ± 3.9 | 63.7 ± 3.8  | 27.5 ± 3.6 |
> | NYT                   | 85.0 ± 2.9 | 46.0 ± 4.0 | 63.8 ± 3.8  | 32.0 ± 3.7 |
> | Wiki                  | 85.2 ± 2.8 | 47.3 ± 4.0 | 67.7 ± 3.7  | 27.5 ± 3.6 |

---

> ### Comment · Reviewer_BRwb · 2025-11-21
> **Reply to rebuttal**
>
> I thank the authors for their detailed reply, and appreciate their effort to improve the quality and clarity of the paper. I read through all other reviews and checked the updated version of the paper.
>
> The updated paper resolves most of the inconsistencies regarding models and hence significantly improves clarity/presentation.
>
> Regarding active filtering: The authors' clarification does make sense. It convinced me that failure to perform active filtering given only two sources allows proving a negative result for more broad/general systems. Hence, I do now believe there is some merit to the task. However, I still think that a broad and general benchmark would need to consider more than two sources, so the generality of the benchmark is still limited.
>
> Given the updated presentation, and since I now see how the active filtering task has some merit, I updated my scores.

---

> > ### Author Response · Authors · 2025-11-21
> >
> > Thanks for your continued engagement and thoughtful feedback!
> > We agree that active filtering experiments with more sources would broaden our results. We've modified our setup and will report back shortly with results for our best-performing models.

---

> > > ### Comment · Reviewer_BRwb · 2025-11-22
> > >
> > > I would just like to clarify that my intent was not to push authors towards new experiments. Especially since I believe scaling up the active filtering task requires some thought and likely broad changes to the writing.

---

> ### Author Response · Authors · 2025-11-22
>
> Our setup can easily be modified to accommodate this feedback in a timely manner. Even using the same exact source documents, it's possible to scale up the task by several times by asking the model to filter out 2, 3, or 4 untrustworthy documents. Note that, unlike "resolving contradictions" documents, "active filtering" documents each contain a unique fact, and so we can easily add more of them in the same context. We strongly suspect all models will have significantly higher hallucination rates on these variants, and they are not overly difficult to incorporate into the paper (in fact we've already started running them). Would these experiments not go a long way towards addressing your concern?

---

> ### Comment · Reviewer_BRwb · 2025-11-22
>
> The experiments are certainly useful; I just don't want to force authors to do more work.
>
> And scaling up the number of documents introduces a new degree of freedom in the ratio between trustworthy and untrustworthy documents. For example, the following settings are both valid and interesting, but measure different things:
>
> 1. N trustworthy sources, 1 untrustworthy one: Does the model focus on a (potentially malicious) untrustworthy source? (~poisoning)
> 2. 1 trustworthy source, N untrustworthy ones: How does model performance degrade with increasingly more untrustworthy sources? A "good" model should stay relatively robust to even many untrustworthy sources.
> 3. N trustworthy sources, M untrustworthy ones: How does performance relate to the ratio of M/N?
> etc.
>
> So while I think that the additional experiments are useful, I also think there is much more to explore than what can reasonably be done in the remaining discussion period. Nevertheless, I of course appreciate the promised additional experiments and think they are useful.

---

> > ### Author Response · Authors · 2025-11-22
> >
> > We're committed to making the paper as good as we reasonably can. Earlier, we were alluding to experiment 2. in your list. Because of the way we've set up the code, it's trivial to implement 1 and 3 as well. It would be too expensive to run the full suite of models on all three variants, but we're more than happy to do a case study of a handful of our best performing models (including GPT 5) across these different settings. We think that will give useful insight.

---

> > > ### Author Response · Authors · 2025-12-04
> > >
> > > We've run the requested active filtering experiments with three model families. The left columns indicate the number of trustworthy and untrustworthy sources in the model's "RAG" context. The score for each model is its hallucination rate averaged across all corresponding combinations of the sources.
> > >
> > > Surprisingly, performance degrades in *all* settings. Models seem to get complacent in settings with more trustworthy sources, allowing untrustworthy sources to slip through more often. Increasing the number of untrustworthy sources also increases hallucination rates, presumably because models have more individual opportunities to blunder (trusting even one untrustworthy source counts as a hallucination in this context).
> > >
> > > **Error rate (%) vs. source counts**
> > >
> > > | Trustworthy | Untrustworthy | GPT-5        | GPT-4o       | Llama 3.3 70B |
> > > |------------:|--------------:|-------------:|-------------:|--------------:|
> > > | 2           | 1             | 9.5 ± 2.4    | 31.0 ± 3.7   | 50.1 ± 4.0    |
> > > | 3           | 1             | 8.8 ± 2.2    | 25.3 ± 3.5   | 40.2 ± 3.9    |
> > > | 1           | 2             | 20.1 ± 3.2   | 76.1 ± 3.4   | 85.4 ± 2.8    |
> > > | 1           | 3             | 29.0 ± 3.6   | 85.9 ± 2.8   | 91.3 ± 2.3    |
> > > | 2           | 2             | 20.5 ± 3.2   | 56.6 ± 4.0   | 72.8 ± 3.6    |
> > > | 3           | 3             | 30.0 ± 3.7   | 63.2 ± 3.9   | 75.4 ± 3.4    |
> > >
> > > | Trustworthy | Untrustworthy | Llama 3.1 8B | Gemma 3 27B  | Gemma 3 4B    |
> > > |------------:|--------------:|-------------:|-------------:|--------------:|
> > > | 2           | 1             | 43.6 ± 4.0   | 55.3 ± 4.0   | 60.0 ± 3.9    |
> > > | 3           | 1             | 37.1 ± 3.9   | 46.3 ± 4.0   | 51.8 ± 4.0    |
> > > | 1           | 2             | 83.0 ± 3.0   | 91.4 ± 2.2   | 91.9 ± 2.2    |
> > > | 1           | 3             | 91.0 ± 2.3   | 96.3 ± 1.5   | 96.1 ± 1.5    |
> > > | 2           | 2             | 70.8 ± 3.6   | 80.0 ± 3.2   | 83.0 ± 3.0    |
> > > | 3           | 3             | 78.2 ± 3.3   | 82.5 ± 3.0   | 85.7 ± 2.8    |

---

### Official Review · Reviewer_aMix · 2025-11-03

**Soundness:** 2
**Presentation:** 2
**Contribution:** 1
**Rating:** 2
**Confidence:** 3

**Summary:**

The work explores the timely subject of detecting the trustworthy sources, especially targeting for the media information. Used three categories of tasks to evaluate the model performances systematically. Results show how the model size, reasoning availablility (i.e., reasoning models or not) impact the model performances and whether the models share similar judgements of the source qualities, and eventually SMeL scores.

**Strengths:**

The topic is timely

Elegantly proposed three categories of the evaluation tasks - ignoring dubious sources, resolving contradictions, and active filtering are all meaningful approaches to deal with such misinformation detection tasks.

Data used for the work are rich - from encyclopedia britannica (academic) to Reddit (a casual internet community forum) to the least trustworthy source (i.e., "unknown").

Well-presented/ summarized results - the performances depending on the model scales, reasoning availability, or source quality assessment to show the overall results. Great visualization for the result analysis.

**Weaknesses:**

The work is too heuristic and missing technical concepts to evaluate the validity of the work.
- the work really depends on the data contents and the currently used data do not seem to have standardized methods to evaluate the validity to replicate the work.
- Not sure about the technical depth of the work. It sounds more like a blog post or report of the model result analysis.

**Questions:**

Would you specify more about the experimental setups?
how did you make three tasks?
- ignoring dubious sources - how did you make the unreliable/irrelevant context, any formats that you use for the datasets to modify the original contents?
- resolving contradictions - how did you make the reliable and perturbed versions of the factual questions? what are the standard filler? What prompts did you use for models to ignore the documents that are not trustworthy?
- active filtering - again it would be nice to share the explicti prompts and why you designed such fixed promtps. etc.

How did you make the SMeL scores?

---

> ### Author Response · Authors · 2025-11-15
>
> We thank the reviewer for their feedback and appreciate that they found the topic timely and our problem framing elegant. We respond to individual points below.
> > The work is too heuristic and missing technical concepts to evaluate the validity of the work.
>
> Could the reviewer elaborate on what they feel is missing from the manuscript? We would be happy to make any necessary changes, but are not sure how to reconcile this criticism with its previously claimed strengths, namely that the task construction is meaningful, that the data is well-curated, and that the results are presented in a cohesive manner.
>
> > the work really depends on the data contents and the currently used data do not seem to have standardized methods to evaluate the validity to replicate the work.
>
> All of the code, data, and prompts used in the experiments and data generation process are open-sourced here: https://anonymous.4open.science/r/smel/README.md, along with instructions for using them. It should be possible to replicate all of our results. Please let us know if there are any issues with this or if there’s anything we’ve forgotten to include!
>
> > Not sure about the technical depth of the work. It sounds more like a blog post or report of the model result analysis.
>
> We would like to emphasize that the paper describes a novel benchmark, not just an analysis of model outputs. To restate our contributions:
>
> We design and release a new benchmark testing an important capability of language models.
> We find that state-of-the-art models fail at surprisingly high rates even on the most trivial variants of this new task, which we feel is of independent interest.
> We include follow-up experiments (prompt, presence or absence of source URL, ordering of sources, & experiments with real data to rule out bias from our synthetic documents) to contextualize our results.
>
> Going further, we respectfully disagree that a “model result analysis” would even be out of scope at this conference; the call for papers includes categories for datasets and benchmarks as well as applications of ML to relevant problems. If the reviewer feels that the style of the presentation is lacking or has additional experiments in mind, we would be happy to work with them to update the manuscript and make any necessary changes.
>
> > ignoring dubious sources - how did you make the unreliable/irrelevant context, any formats that you use for the datasets to modify the original contents?
>
> All three SMeL Test tasks draw from the same bank of synthetic documents (apart from the real documents we also test on), whose generation process is described in Section 2.1 and Appendix B. Please let us know if including more of Appendix B in the main text would be useful.
>
> > resolving contradictions - how did you make the reliable and perturbed versions of the factual questions? what are the standard filler? What prompts did you use for models to ignore the documents that are not trustworthy?
>
> Please see above. For prompts, see Appendix F.3. In all cases, the models are instructed to ignore documents that are not trustworthy, apart from when we explicitly remove this warning to test the model’s reliance on a specific type of prompt, which we describe in the paper.
>
> > How did you make the SMeL scores?
>
> To make the overall SMeL scores in Table 2, we averaged each model’s results across document pairings on the three constituent tasks (i.e., synthesis of Figures 2, 3, and 4).
>
>
> Thank you in advance for your time. Please let us know if we have missed any of the reviewer’s concerns or if there’s anything else standing in the way of a stronger endorsement.

---

### Meta-Review · Area_Chair_7QAA · 2025-12-29

**Summary:**

The paper introduces the Synthetic Media Literacy Test (SMeL), a benchmark designed to evaluate how well LLM filter out untrustworthy or fictional information across three tasks: ignoring dubious sources, resolving contradictions, and active filtering.

During the rebuttal phase, the authors addressed several reviewer concerns, such as expanding the "active filtering" task to include more documents, clarifying model versions (e.g., GPT-4o and Gemini 2.5 Pro), and improving the presentation of results. While some reviewers (aokR and BRwb) found the benchmark timely and well-executed, others (aMix and De3v) remained highly critical of the work's fundamental approach. For example, the testing is primarily based on some predefined "heuristic rules" (such as Britannica being good, Fanfiction being bad). In the real world, however, trustworthiness is extremely complex, and the paper’s treatment of "real-world complexity" is considered overly simplistic.

**Reviewer Concerns:**

Significant concerns remain regarding the technical depth and heuristic nature of the benchmark. Reviewers aMix and De3v noted that the work lacks "technical concepts" and that the results depend too heavily on specific data contents rather than generalized principles. Furthermore, there are unresolved criticisms regarding the definition of "ground truth" (fictional vs. factual).

**Reviewer Scores:**

The final scores are split: 8 (aokR), 6 (BRwb - raised from 4), 2 (aMix), and 2 (De3v). While one reviewer was strongly in favor and another was satisfied by the rebuttal, two reviewers maintained their strong rejection.

---

### Decision · Program_Chairs · 2026-01-26

Reject